# Preoperative myocardial expression of E3 ubiquitin ligases in aortic stenosis patients undergoing valve replacement and their association to postoperative hypertrophy

Fábio Trindade[1,2], Francisca Saraiva[2], Simon Keane[3], Adelino Leite-Moreira[2], Rui Vitorino[1,2], Homa Tajsharghi[3], Inês Falcão-Pires[2]*

1 Department of Medical Sciences, iBiMED–Institute of Biomedicine, University of Aveiro, Aveiro, Portugal, 2 Department of Surgery and Physiology, UnIC—Cardiovascular Research and Development Centre, Faculty of Medicine, University of Porto, Porto, Portugal, 3 Division Biomedicine, School of Health Sciences, University of Skövde, Skövde, Sweden

* ipires@med.up.pt

**Data Availability Statement:** All relevant data are within the manuscript and its Supporting Information files.

## Abstract

Currently, aortic valve replacement is the only treatment capable of relieving left ventricle pressure overload in patients with severe aortic stenosis. It aims to improve cardiac function and revert hypertrophy, by triggering myocardial reverse remodeling. Despite immediately relieving afterload, reverse remodeling turns out to be extremely variable. Among other factors, the extent of reverse remodeling may depend on how well ubiquitin-proteasome system tackle hypertrophy. Therefore, we assessed tagged ubiquitin and ubiquitin ligases in the left ventricle collected from patients undergoing valve replacement and tested their association to the degree of reverse remodeling. Patients were classified according to the regression of left ventricle mass (ΔLVM) and assigned to complete (ΔLVM≥15%) or incomplete (ΔLVM≤5%) reverse remodeling groups. No direct inter-group differences were observed. Nevertheless, correlation analysis supports a fundamental role of the ubiquitin-proteasome system during reverse remodeling. Indeed, total protein ubiquitination was associated to hypertrophic indexes such as interventricular septal thickness ($r = 0.55$, $p = 0.03$) and posterior wall thickness ($r = 0.65$, $p = 0.009$). No significant correlations were observed for Muscle Ring Finger 3. Surprisingly, though, higher levels of atrogin-1 were associated to postoperative interventricular septal thickness ($r = 0.71$, $p = 0.005$). In turn, Muscle Ring Finger 1 correlated negatively with this postoperative hypertrophy marker ($r = -0.68$, $p = 0.005$), suggesting a cardioprotective role during reverse remodeling. No significant correlations were found with left ventricle mass regression, although a trend for a negative association between the ligase Murine Double Minute 2 and mass regression ($r = -0.44$, $p = 0.10$) was found. Animal studies will be necessary to understand whether this ligase is protective or detrimental. Herein, we show, for the first time, an association between the preoperative myocardial levels of ubiquitin ligases and postoperative hypertrophy, highlighting the therapeutic potential of targeting ubiquitin ligases in incomplete reverse remodeling.

**Funding:** The authors thank Portuguese Foundation for Science and Technology (FCT), European Union, Quadro de Referência Estratégico Nacional (QREN), Fundo Europeu de Desenvolvimento Regional (FEDER) and Programa Operacional de Competitividade (COMPETE) for funding iBiMED (UID/BIM/04501/2020 and POCI-01-0145-FEDER-007628) and UnIC (UIDB/00051/2020 and UIDP/00051/2020) research units. This project is supported by FEDER through COMPETE 2020 – Programa Operacional Competitividade E Internacionalização (POCI), the project DOCNET (NORTE-01-0145-FEDER-000003), supported by Norte Portugal regional operational programme (NORTE 2020), under the Portugal 2020 partnership agreement, through the European Regional Development Fund (ERDF), the project NETDIAMOND (POCI-01-0145-FEDER-016385, SAICTPAC/0047/2015), supported by European Structural And Investment Funds, Lisbon's regional operational program 2020. HT is supported by the European Union's Seventh Framework Programme for research, technological development and demonstration under grant agreement no. 608473 and the Swedish Research Council. RV and FT are supported by IF/00286/2015 and SFRH/BD/111633/2015 fellowship grants, respectively. FS is supported by Universidade do Porto/FMUP and FSE-Fundo Social Europeu, NORTE 2020-Programa Operacional Regional do Norte, NORTE-08-5369-FSE-000024-Programas Doutorais. The authors would like also to address a special thanks to COST Action BM1307/PROTEOSTASIS for funding this work and granting FT with a short-term scientific mission stipend. The funders had no role in study design, data collection and analysis, decision to publish, or preparation of the manuscript.

**Competing interests:** The authors have declared that no competing interests exist.

## Introduction

Aortic stenosis (AS) is the most common valvular heart disease in developed countries. AS is also a serious condition being associated with higher risk of adverse events such as myocardial infarction, stroke and death [1]. The severity of AS arises from the progressive thickening and narrowing of the valve. These structural changes impose chronic pressure overload that gradually increases the wall stress in left ventricle (LV). To restore or normalize wall stress, LV cardiomyocytes undergo hypertrophy. These factors collectively promote structural and functional deterioration. In untreated cases, maladaptive remodeling may converge into heart failure [2]. Therefore, AS patients require timely intervention before the myocardium can no longer reverse its pathological state. Currently, surgical aortic valve replacement (AVR) remains as the standard therapeutic option [3].

Following AVR, patients experience an immediate relief of LV afterload, which triggers myocardial reverse remodeling (RR). This response is mainly characterized by regression of LV hypertrophy, restoration of chamber shape and a concomitant improvement of LV function [4–6]. However, the extent of RR is extremely variable. Some patients show signs of incomplete RR while others may attain complete structural and functional recovery. Systemic hypertension [7–10], diabetes mellitus [8], gender [9,11], fibrosis [12], patient-prosthesis mismatch [13] in addition to aging [14], have all been suggested to explain such a variable response.

To keep proteostasis, cardiomyocytes are harnessed with two systems, the ubiquitin-proteasome system (UPS) and the autophagy pathway. These are responsible to signal and degrade worn proteins and recycle sarcomeric proteins, the main determinants of cardiac mass [15]. Particularly in what regards to UPS, the activity of E3 ubiquitin ligases is of paramount importance, as these confer substrate-specificity (reviewed in [16,17]). A growing body of evidence shows the relationship between these enzymes and cardiac disease, particularly hypertrophy. For instance, patients with compound Muscle Ring Finger (MuRF)1 deficiency and a deleterious MuRF3 missense mutation develop cardiomyopathy [18]. Similarly, MuRF1/MuRF3 double knockout mice develop extreme muscle hypertrophy [19]. Mice overexpressing atrogin-1 in the heart display a blunted hypertrophy phenotype and decreased apoptosis when subjected to pressure overload [20]. Finally, Murine Double Minute 2 (MDM2) overexpression in cardiomyocytes stimulated with pro-hypertrophic $\alpha$-agonists (phenylephrine and endothelin-1) mitigates hypertrophy and inhibits the fetal gene program [21].

Thus, since hypertrophy reversal is a necessary requisite for complete RR and E3 ubiquitin ligases have been implicated in the regulation of hypertrophy, we hypothesized that preoperative LV levels of total protein ubiquitination and of the ligases MuRF1, MuRF3, atrogin-1 and MDM2 are associated with the degree of RR. Thus, we assessed the levels of the latter UPS elements in the LV of patients undergoing AVR, compared between patients with complete and incomplete RR and tested their association with postoperative clinical parameters, including LV mass regression.

## Material and methods

### Study design, patients selection and clinical characterization

AS patients were selected based on retrospective clinical data, and the respective myocardial samples were gathered from the local biobank. The local ethics committee approved the study protocol, and written informed consent was obtained from all patients. This study complies with the Declaration of Helsinki.

Only patients undergoing AVR with clinical predominance of AS, without severe aortic insufficiency or severe forms of other extra-aortic valve diseases and with no more than one stenotic coronary vessel were selected. Clinical evaluation of AS severity and myocardial structure and function was based on transthoracic echocardiography. Peak aortic valve velocity (Peak Ao), mean aortic transvalvular pressure gradient and indexed aortic valve area (AVAi) were derived from Doppler echocardiography. Mean pressure gradient was obtained with the modified Bernoulli equation and AVAi with the standard continuity equation. In turn, LV end-diastolic dimension (LVEDD), LV posterior wall thickness (PWT) and interventricular septal thickness (IVST) were derived from 2D-echocardiograms during diastole. Relative wall thickness (RWT) was calculated as $2 \times PWT/LVEDD$. Correct orientation of imaging planes, cardiac chambers dimension and function measurements were performed according to the EAE/ASE recommendations [22].

LV mass (LVM) was estimated according to the joint recommendations of the EAE and ASE using Devereux's formula for ASE measurements in diastole: LV mass = $0.8 \times (1.04 \times$ ([LV internal dimension + PWT + IVST]$^3$ –[LV internal dimension]$^3$) + 0.6 g. LVM index (LVMi) was calculated according to the recent recommendations for cardiac chamber quantification [22]. LVMi greater than 115 g/m$^2$ in men and greater than 95 g/m$^2$ in women were considered indicative of hypertrophy.

The clinical and demographical of the final population (n = 15) is summarized in **Table 1**. All patients enrolled in this study were free of dilated or hypertrophic cardiomyopathies. Upon preoperative echocardiographic assessment, patients presented with a mean preoperative RWT >0.42, indicative of a more concentric remodeling, typical of pressure overload-induced myocardial remodeling. These patients showed LVEF >50% (n = 12) or had clinical indication of normal/good systolic function (n = 2). Only one patient showed a borderline LVEF (42%). Hence, the evaluation of RR was based on echocardiographic assessment of LV hypertrophy. LVM regression (%) was defined as the difference between pre- and postoperative LVMi. Patients with LVM regression (ΔLVM) ≥15% were included in the complete RR (cRR) group, while those with LVM regression ≤5% were integrated into the incomplete RR (iRR) group. Complete and incomplete RR groups were not different regarding risk factors such as age, hypertension, obesity, diabetes and smoking. All patients displayed severe or were borderline between moderate to severe AS, according to the ESC/AHA guidelines [23].

## Sample collection and processing

During AVR, LV myocardial biopsies were collected and immediately frozen at -80˚C. LV biopsy material consisted of endomyocardial tissue resected from the LV outflow tract (Morrow procedure) because of concomitant LV outflow tract hypertrophy. The myocardial tissue of the AS patients and from a healthy donor heart (used as technical control) were subjected to the same homogenization and protein extraction protocol. Briefly, biopsies were homogenized in RIPA buffer (40 μL/mg tissue; Thermo Scientific™ cat #89900) supplemented with a cocktail of protease inhibitors (1:100; Sigma-Aldrich® cat #8340) by mechanical disruption using a disposable tissue homogenizer. Protein-rich, debris-free supernatants were recovered after centrifugation for 30 min at 13,000 × *g* (4˚C), aliquoted and stored in liquid nitrogen. Protein concentration was estimated with a BCA assay kit, using bovine serum albumin as standard.

## Western blot experiments

Protein extracts (10 μg) were separated by SDS-PAGE on 8–16% or 4–20% (for atrogin-1 only) gradient TGX stain-free gels, essentially as described by Laemmli [24], and gels were scanned on a ChemiDoc MP imager (BioRad). Next, proteins were transferred to low

**Table 1. Clinical data of the study population.**

| Parameters | | | |
|---|---|---|---|
| **Degree of Reverse Remodeling** | **Complete (ΔLVM≥15%)** | **Incomplete (ΔLVM≤5%)** | ***p*** |
| **Demographics** | | | |
| N | 8 | 7 | ns |
| Age | 68.7±3.2 | 68.8±3.4 | ns |
| Gender (male:female) | 4:3 | 1:6 | ns |
| BMI (kg/m$^2$) | 30.2±9.2 | 33.6±6.3 | ns |
| Obesity (n) | 3 | 5 | ns |
| Hypertension (n) | 5 | 5 | ns |
| Diabetes mellitus (n) | 3 | 2 | ns |
| CAD (≤1 vessel) (n) | 1 | 2 | ns |
| Smoking history (n) | 4 | 1 | ns |
| COPD (n) | 2 | 1 | ns |
| Mild-to-moderate aortic insufficiency (n) | 8 | 5 | ns |
| Mild-to-moderate mitral stenosis (n) | 0 | 0 | ns |
| Mild-to-moderate mitral insufficiency (n) | 7 | 5 | ns |
| Mild-to-moderate tricuspid insufficiency (n) | 6 | 4 | ns |
| **Preoperative parameters** | | | |
| AVAi, cm$^2$/m$^2$ | 0.44±0.04 | 0.44±0.04 | ns |
| Peak Ao, m/s | 4.7±0.6 | 4.5±0.5 | ns |
| Max ATPG, mmHg | 89.6±22.1 | 80.6±17.9 | ns |
| Mean ATPG, mmHg | 56.3±13.5 | 51.0±11.8 | ns |
| LVEDD, cm | 5.2±0.2 | 4.8±0.5 | ns |
| IVST, cm | 1.4±0.2 | 1.3±0.2 | ns |
| PWT, cm | 1.3±0.1 | 1.2±0.2 | ns |
| RWT | 0.50±0.05 | 0.50±0.09 | ns |
| LVMi, g/m$^2$ | 164.9±32.0 | 130.1±26.8 | * |
| **Postoperative parameters** | | | |
| LVEDD, cm | 4.8±0.4 | 5.0±0.5 | ns |
| IVST, cm | 1.3±0.2 | 1.4±0.1 | ns |
| PWT, cm | 1.0±0.2 | 1.1±0.1 | a |
| RWT | 0.41±0.07 | 0.45±0.06 | ns |
| LVMi, g/m$^2$ | 110.6±23.8 | 139.6±27.9 | * |
| ΔLVM, % | 32.9±7.2 | -7.9±9.9 | b |

Abbreviations: ATPG: aortic transvalvular pressure gradient; AVAi: aortic valve area, indexed to body surface area; BMI: body mass index; CAD: coronary artery disease; COPD: chronic obstructive pulmonary disease; IVST: interventricular septal thickness; LVEDD: left ventricle end-diastolic dimension; LVMi: left ventricle mass, indexed to body surface area; Peak Ao: peak aortic valve velocity; PWT: posterior wall thickness; RWT: relative wall thickness; ΔLVM: left ventricle mass regression.

* $p < 0.05$ (unpaired two-tailed t-test)

[a] $p = 0.06$

[b] Independent variable

fluorescence polyvinylidene difluoride membranes using BioRad's Trans-Blot® Turbo™ system with a customized setting (2.5 A, 25 V, 8 min). Transfer efficiency was evaluated by scanning the membranes and rescanning the gels. All membranes were blocked with 1% bovine serum albumin for 30 min and incubated overnight (4°C) with the primary antibodies. Following several washes with TBS-T, membranes were incubated for one hour with the respective secondary antibodies and washed again with TBS-T. **Table 2** compiles all the antibodies used.

**Table 2. List of antibodies used in western blot.**

| Protein target | UniProt ID and Gene Name | Primary antibody | Secondary antibody |
|---|---|---|---|
| Ubiquitin | Not applying—the antibody targets ubiquitin and polyubiquitin chains | Mouse mAb 1:500; sc-166553 | Peroxidase-AffiniPure Goat pAb anti-mouse 1:10000; Jackson Immunoresearch lab. 115-035-062 |
| Muscle Ring Finger 1 | Q969Q1 TRIM63 | Goat pAb 1:200; ab4125 | HRP-linked Donkey pAb anti-goat 1:10000; sc-2020 |
| Muscle Ring Finger 3 | Q9BYV2 TRIM54 | Goat pAb 1:500; ab4351 | HRP-linked Donkey pAb anti-goat 1:10000; sc-2020 |
| Atrogin-1 | Q969P5 FBXO32 | Mouse mAb 1:500; sc-166806 | DyLight 800-conjugated Goat pAb anti-mouse; Bio-Rad STAR117D800GA |
| Murine Double Minute 2 | Q00987 MDM2 | Mouse mAb 1:500; sc-965 | Peroxidase-AffiniPure Goat pAb anti-mouse 1:10000; Jackson Immunoresearch lab. 115-035-062 |

Abbreviations: HRP: horseradish peroxidase; mAb: monoclonal antibody; pAb: polyclonal antibody

Chemiluminescence-based detection was carried out with an enhanced chemiluminescence system kit (Clarity Max^TM Western ECL, BioRad), using ChemiDoc MP imaging system. Fluorescence-based detection was performed with the same imaging system. Acknowledging the findings of Curi *et al.* [25], OD normalization was done in relation to total protein levels. Although, instead of the standard Ponceau S staining, protein's signal OD was normalized to total lane OD after transfer. This was possible due to the utilization of the stain-free technology, with greater linear dynamic range than the typical Ponceau S method, as stated by the manufacturer (Bio-Rad). To compare samples in different blots, the OD signal was also normalized to the technical control sample. Blot scans were analyzed with ImageLab 5.1 software (Bio-Rad).

## Statistical analysis

Categorical clinical data are presented as absolute frequencies. Fisher's exact test was applied to detect differences between both groups (cRR and iRR). Continuous demographical and molecular expression data are presented as mean ± standard deviation. The normality of the distribution was tested by the D'Agostino & Pearson omnibus method. The differences between groups were tested with the unpaired two-tailed t-test, if the variables were normally distributed, or with the Mann-Whitney test, if otherwise. Correlation between clinical parameters and the relative expression of the proteins was evaluated with Pearson's test if data presented a normal distribution and with Spearman's test, if otherwise. All statistical tests were done with GraphPad Prism 6 and, in any case, $p < 0.05$ was considered significant.

## Results

### Characterization of the preoperative levels of ubiquitin and of hypertrophy-associated ubiquitin ligases in the left ventricle

Protein recycling by the proteasome requires prior ubiquitination. Therefore, we hypothesized that cRR patients present higher preoperative levels of total protein ubiquitination. Immunodetection of ubiquitinated proteins, however, showed no significant difference in total ubiquitin levels between the two groups (**Fig 1A and 1F**). Perhaps, ubiquitination of rather specific substrates may underscore the difference between these groups. Moreover, an incomplete response may also arise from dysregulation of the E3 ubiquitin ligases involved in marking proteins towards proteasome turnover. In this sense, the relative expression of three muscle-specific ubiquitin ligases, namely MuRF1 (**Fig 1B and 1G**), MuRF3 (**Fig 1C and 1H**) and

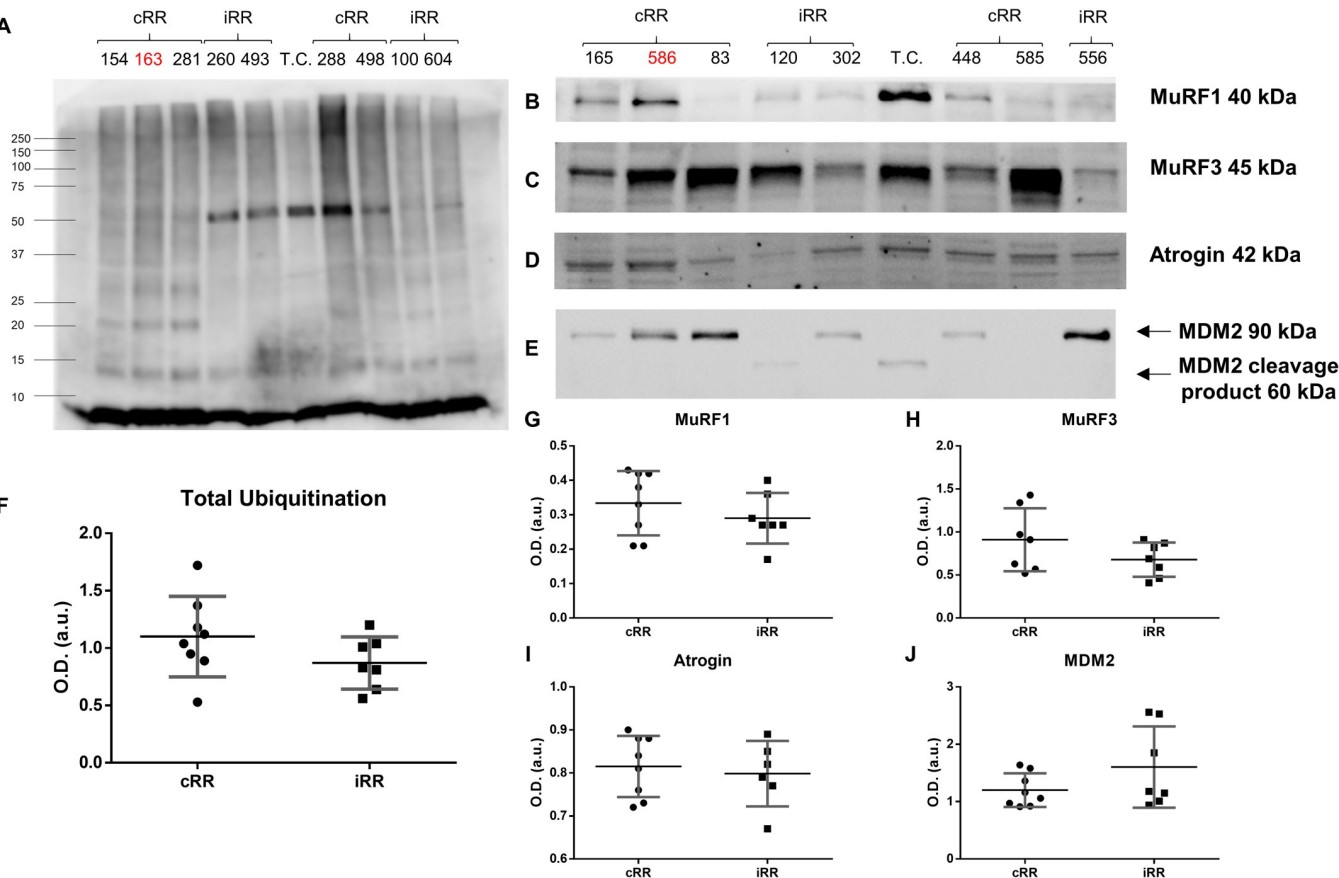

**Fig 1. Myocardial ubiquitination profile and ubiquitin ligases expression in patients with complete or incomplete reverse remodeling.** Representative western blot scans are depicted for ubiquitin-tagged proteins (A) and for the E3 ubiquitin ligases Muscle Ring Finger (MuRF) 1 (B), MuRF3 (C), atrogin-1 (D) and Murine Double Minute 2 (MDM2) (E). Additional blots can be found in S1 Fig. In the latter, two bands are identified with arrows, referring to intact MDM2 (upper arrow) and its cleavage product (lower arrow), as reported in the datasheet. Quantification was based on the intact band only. The respective optical density-based semi-quantification is shown in (F), (G), (H), (I) and (J). T.C. designates the technical control. Samples 163, 586 and 603 are marked in red because these were excluded from the study *a posteriori* (aortic or mitral insufficiency was found to be more or as severe as aortic stenosis). cRR identifies patients with complete reverse remodeling and iRR denotes patients with incomplete reverse remodeling.

atrogin-1 (**Fig 1D and 1I**), which are known to stimulate atrophy or blunt hypertrophy [18–20] were assessed. Additionally, the ubiquitous MDM2 was also measured (**Fig 1E and 1J**), due to previously reported cardioprotective effects in a different pathological setting [21]. Surprisingly, no differences were found for these UPS players, despite the apparently higher levels of MuRF3 (1.3-fold, *p* = 0.17) in cRR patients, which did not reach statistical significance.

## Correlation analysis

Considering that most of the clinical parameters measured are continuous variables, correlation analysis was performed to foresee potential associations between preoperative levels of ubiquitin and its ligases with patients' outcomes. Comparison between clinical parameters showed that patients with higher preoperative LVMi displayed a higher tendency to LVM regression, as demonstrated by a positive association with ΔLVM (**Table 3**). Besides, LVM regression should rely mainly on the normalization of the PWT, since this parameter was out of the three echocardiographic indexes of chamber geometry (LVEDD, IVST and PWT), the one more closely associated to ΔLVM (r = 0.49, *p* = 0.07). This is expected considering the

**Table 3. Correlation analysis between clinical and molecular data.**

| Clinical Parameters | ΔLVM | Molecular Data | | | | |
| --- | --- | --- | --- | --- | --- | --- |
| | | Ubiquitin[a] | MuRF1 | MuRF3 | Atrogin-1 | MDM2[b] |
| AVAi | | r = -0.63 $p = 0.03$ | r = 0.59 $p = 0.04$ | r = 0.54 $p = 0.09$ | | |
| IVST (preop.) | | r = 0.55 $p = 0.03$ | | | | |
| IVST (postop.) | | | | | r = 0.71 $p = 0.005$ | |
| PWT (preop.) | | r = 0.65 $p = 0.009$ | | | r = 0.49 $p = 0.08$ | |
| PWT[b] (postop.) | | | r = -0.68 $p = 0.005$ | | | |
| RWT (preop.) | | | r = -0.56 $p = 0.03$ | | | |
| RWT (postop.) | | | | | | |
| LVMi (preop.) | r = 0.55 $p = 0.04$ | | | | r = 0.70 $p = 0.006$ | |
| LVMi (postop.) | r = -0.56 $p = 0.03$ | | | | | |
| | **ΔLVM** | | | | | r = -0.44 $p = 0.10$ |

No correlations were found between ubiquitin or its ligases and age, body mass index, peak aortic valve velocity or to mean/maximal aortic transvalvular pressure gradients.

Marginally significant correlations ($p \leq 0.10$) are shown in grey.

Abbreviations: MuRF; Muscle Ring Finger Protein; MDM2: Murine Double Minute 2; AVAi: indexed aortic valve area; LVEDD: left ventricle end-diastolic dimension; IVST: interventricular septal thickness; PWT: posterior wall thickness; LVMi; indexed left ventricle mass, indexed to body surface area; ΔLVM: left ventricle mass variation.

[a] Total protein ubiquitination

[b] Non-normal distribution; Spearman's test was used instead

pattern of concentric remodeling commonly found in AS patients [26]. Furthermore, AVAi correlated negatively with preoperative IVST (r = -0.69, $p = 0.01$) and both Peak Ao and the maximal transvalvular pressure gradient correlated positively with LVMi (r = 0.55, $p = 0.03$), showing that the magnitude of hypertrophy is associated with AS severity. Altogether, these results reassure the consistency of clinical data.

Also presented in **Table 3** are the correlations between the levels of protein ubiquitination and ligases with several clinical parameters. Total levels of ubiquitination were found inversely correlated with AVAi (r = -0.63, $p = 0.03$). Significant positive correlations were also observed between ubiquitin and preoperative IVST (r = 0.55, $p = 0.03$) and PWT (r = 0.65, $p = 0.009$), both markers of hypertrophy. As opposed to ubiquitin, MuRF1 was positively associated to AVAi (r = 0.59, $p = 0.04$). MuRF1 also showed a negative association with preoperative RWT (r = -0.56, $p = 0.03$). Somehow unexpected, though, a strong positive association of atrogin-1 with preoperative LVMi (r = 0.70, $p = 0.006$) is evident. Since it is unethical to collect LV from patients after AVR, the variation of total protein ubiquitination and of ligases during RR cannot be evaluated. Still, their association with several post-clinical parameters may be analyzed. Regarding MuRF1, adding to its inverse association with preoperative RWT, this ligase showed a negative association to postoperative PWT (r = -0.68, $p = 0.005$). Atrogin, in turn, was positively associated with postoperative IVST (r = 0.71, $p = 0.005$). No correlations were found between MuRF3 and any of the pre- or postoperative parameters analyzed. Concerning LVM

regression, the most important parameter to determine the success of AVR in patients' outcome, out of the five UPS elements analyzed, MDM2 showed the strongest association (r = -0.44), approaching significance ($p$ = 0.10). Curiously, MDM2, unlike the remaining ligases, is the only showing an inverse relationship with ΔLVM.

## Discussion

Despite directly addressing AS' superimposed pressure overload, AVR may not always be an effective treatment. In fact, some patients still exhibit LV hypertrophy after surgery and demand further clinical attention and care. In this regard, identifying potential novel therapeutic targets to maximize the outcome of patients displaying iRR can be helpful. Based on the premise that UPS plays a central role in sarcomere recycling and is required for the management of hypertrophy [15], the association between the preoperative levels of ubiquitin and of its ligases with the degree of RR was tested. When comparing the two study groups, it was clear that no significant differences were present. This may be explained by the high standard deviation of the protein levels, which, in turn, should reflect the high variation in LVM regression. For that reason, and acknowledging that clinical parameters were continuous variables, correlation analysis was performed, in an effort to uncover associations between UPS elements and markers of AS severity, hypertrophy and RR.

Proteasome-based degradation of proteins relies on the serial addition of ubiquitin chains [16]. Thus, the global ubiquitination profile in the LV was studied. Interestingly, an association between the amount of ubiquitinated proteins and the severity of AS was observed, as patients with lower valve area (AVAi) presented higher ubiquitin expression. Besides, positive correlations between hypertrophy (preoperative IVST and PWT) and total levels of ubiquitinated proteins were found. Therefore, it is tempting to suggest that the more stenotic the valves are, the higher will be the magnitude of LV hypertrophy and, consequently, the levels of ubiquitinated proteins. Perhaps, depending on the specificity of the ubiquitination, conferred by E3 ligases, and the (dys)regulation of E3 ligases itself, such proteins may or may not be recycled on the proteasome. This may be one of the reasons why some patients present complete and others incomplete RR.

Many ubiquitin ligases have been studied in the heart and reported to regulate cell trophic response [16,17]. Since RR relies heavily on the reversal of hypertrophy, the levels of post-natal muscle-specific ubiquitin ligases, MuRF1, MuRF3 and atrogin-1, were studied. MuRF1 has been reported to regulate hypertrophy by downregulation of the calcineurin A-nuclear factor of activated T cells axis, through ubiquitination and consequent proteasomal delivery [27]. The relevance of MuRF1 and of MuRF3 is also demonstrated by extreme hypertrophy in a MuRF1 and MuRF3 double knockout model [19]. MuRF1 was found negatively associated with postoperative PWT. Thus, although, not directly associated with LV regression, MuRF1 may be important for the normalization of LV chamber geometry, by limiting LV's posterior wall thickening. This is corroborated by a previous study showing increased MuRF1 levels in atrophied LV samples collected after implantation of LVADs [28]. Intriguingly, though, MuRF1 levels were lower in patients with narrower aortic valves (positive correlation with AVAi), suggesting a possible downregulation with increased AS severity that requires confirmation. The process of sarcomere recycling has also been associated with MuRF3 activity. MuRF3-knockout mice show hypertrophic cardiomyopathy associated with subsarcolemmal accumulation of myosin [19]. However, in this setting, it seems that MuRF3 activity may not be as important as the remaining ligases, since, no correlation was found with any of the clinical parameters. Atrogin-1 expression was also hypothesized to be an important factor in determining LVM regression after AVR. The rationale is supported by, for instance, reduced

hypertrophy and apoptosis in atrogin-1 overexpressing mice subjected to myocardial pressure overload [20]. Surprisingly, though, atrogin-1 showed a strong positive correlation with preoperative LVMi and with postoperative IVST. If, at one hand, the association with LVMi suggests that atrogin-1 expression is a response to growing cardiomyocytes; at the other hand, the association to postoperative IVST (a hypertrophy marker) suggests that such induction might not be beneficial. In fact, some reports challenge the idea of atrogin-1 being cardioprotective in AS-induced myocardial remodeling. While the expression of this enzyme rises in response to pressure overload-induced hypertrophy [29], atrogin also shows a profibrotic role. This is corroborated by the fact that atrogin-1 knockout mice show smaller hearts and reduced interstitial fibrosis when subjected to aortic constriction-induced pressure overload [30]. Mechanistically, this is explained by lower activation of the transcription factor nuclear factor-κB, because transgenic mice presented higher levels of its cytoplasmic inhibitor (IκBα). Given the contradictory findings [20,29,30], more research is required to understand the definitive role of atrogin-1 in cardiac hypertrophy induced by AS.

Although MDM2 (a.k.a. HDM2, double minute 2 protein) is traditionally associated with tumorigenesis, due to its p53 suppressing activities, it also targets sarcomeric proteins for degradation, such as telethonin [31]. Previous experiments have shown a cardioprotective role of MDM2. Overexpression of MDM2 efficiently mitigated the development of hypertrophy and fetal gene program activation in cardiomyocytes stimulated *in vitro* with phenylephrine and endothelin-1 [21]. Others reported that MDM2 increases in response to treatment with insulin-like growth factor-1, thus protecting from stretch-mediated apoptosis, through the arrest of p53 function [32]. Finally, ablation of cardiac MDM2 resulted in concentric hypertrophy, wall thickening and interstitial fibrosis. This was translated into lower fractional shortening and higher mortality in a Mdm2-knockout mouse model [33]. Therefore, despite the ubiquitous expression, MDM2 expression was also hypothesized to affect the degree of myocardial RR after AVR. Even though only a trend, MDM2 was of all ubiquitin ligases, the one showing the closest association to LVM regression (r = -0.44, $p$ = 0.1). Curiously, though, a negative relationship was observed, suggesting that patients with higher levels of MDM2 are at higher risk of post-AVR iRR. One hypothesis to explain this observation is that, despite being initially protective, persistent activation can be detrimental. In fact, MDM2 inhibits p53 transcriptional activity and promotes its degradation through direct ubiquitination and targeting to proteasome, preventing apoptosis [34]. Notwithstanding, other pathways can be activated by MDM2 and negatively regulate RR. Upregulation of MDM2 leads to sustained inflammation and fibrosis also through the degradation of the IκBα [35], favoring hypertrophy as well [36]. Thus, the present data and the existing evidence together suggest that long-term activation of MDM2 may not be beneficial for myocardial recovery. Anyhow, since only a trend for an association between MDM2 and LVM regression was observed, a screen in a bigger population will be necessary. If confirmed, the definitive role of MDM2, in the context of RR, should be elucidated in animal models of Mdm2 overexpression/ablation, considering that current evidence is conflicting.

## Conclusions

In summary, studying human cardiac tissue is challenging. Heart biopsies are only available upon surgery, and the collected material is scarce but highly valuable. The biggest limitation of the present work was the impossibility to perform molecular analysis on LV after surgery. Consequently, the assessment of myocardial recovery was limited to echocardiographic examination, and no causal relationships could be determined between variation of UPS players and the incompleteness of RR. Nonetheless, by assessing proteins governing proteostasis in heart muscle and testing their association with clinical parameters some ubiquitin ligases arose as

important players in the process of RR, meriting further scrutiny. This is the case for MuRF1, herein showing a negative correlation with a hypertrophy marker, the postoperative PWT. MuRF1 prognostic role requires tests in larger cohorts and, if possible, in a longitudinal manner. Also worthy of further focus of research is MDM2, the ubiquitin ligase with the highest association with LVM regression. Current evidence points to a rather adverse role of MDM2 in the progression of RR. MDM2 may, thus, become an important therapeutic target, though, only by performing animal studies a definitive conclusion will be achieved.

## Supporting information

**S1 Fig. Myocardial ubiquitination profile and ubiquitin ligases expression in patients with complete or incomplete reverse remodeling.** Remaining western blot scans are depicted for ubiquitin-tagged proteins (A) and for the E3 ubiquitin ligases Muscle Ring Finger (MuRF) 1 (B), MuRF3 (C), atrogin-1 (D) and Murine Double Minute 2 (MDM2) (E). The respective optical density-based semi-quantification is shown in Fig 1(F)–1(J). T.C. designates the technical control. Samples 163, 586 and 603 are marked in red because these were excluded from the study *a posteriori* (aortic or mitral insufficiency was found to be more or as severe as aortic stenosis). 'M' means missing. cRR identifies patients with complete reverse remodeling and iRR denotes patients with incomplete reverse remodeling.
(TIF)

**S1 Raw images.**
(PDF)

## Author Contributions

**Conceptualization:** Fábio Trindade, Rui Vitorino, Homa Tajsharghi, Inês Falcão-Pires.

**Data curation:** Francisca Saraiva.

**Formal analysis:** Fábio Trindade, Francisca Saraiva, Inês Falcão-Pires.

**Funding acquisition:** Adelino Leite-Moreira, Rui Vitorino, Homa Tajsharghi, Inês Falcão-Pires.

**Investigation:** Fábio Trindade.

**Methodology:** Fábio Trindade, Simon Keane.

**Project administration:** Rui Vitorino, Homa Tajsharghi.

**Resources:** Adelino Leite-Moreira, Homa Tajsharghi, Inês Falcão-Pires.

**Supervision:** Adelino Leite-Moreira, Rui Vitorino, Homa Tajsharghi, Inês Falcão-Pires.

**Validation:** Fábio Trindade.

**Visualization:** Fábio Trindade.

**Writing – original draft:** Fábio Trindade.

**Writing – review & editing:** Fábio Trindade, Francisca Saraiva, Simon Keane, Adelino Leite-Moreira, Rui Vitorino, Homa Tajsharghi, Inês Falcão-Pires.

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
