## [Decision Letter · Decision Letter 0]

26 May 2020

PONE-D-20-03272

Preoperative myocardial expression of E3 ubiquitin ligases in aortic stenosis patients undergoing valve replacement and their association to postoperative hypertrophy

PLOS ONE

Dear Dr. Falcão-Pires,

Thank you for submitting your manuscript to PLOS ONE. After careful consideration, we feel that it has merit but does not fully meet PLOS ONE’s publication criteria as it currently stands. Therefore, we invite you to submit a revised version of the manuscript that addresses the points raised during the review process.

We look forward to receiving your revised manuscript.

Kind regards,

Cécile Oury

Academic Editor

PLOS ONE

Journal Requirements:

Reviewers' comments:

Reviewer's Responses to Questions

**Comments to the Author**

1. Is the manuscript technically sound, and do the data support the conclusions?

Reviewer #1: Yes

2. Has the statistical analysis been performed appropriately and rigorously? 

Reviewer #1: Yes

3. Have the authors made all data underlying the findings in their manuscript fully available?

Reviewer #1: Yes

4. Is the manuscript presented in an intelligible fashion and written in standard English?

Reviewer #1: Yes

5. Review Comments to the Author

Reviewer #1: The authors set out to identify factors that affect the degree of reverse remodeling following aortic valve replacement. The authors specifically focus their analysis on the ubiquitin-proteasome system (UPS) by measuring preoperative levels of total ubiquitination and several E3 ubiquitin ligases. Pre and postoperative echocardiography data were used to segregate patients into incomplete or complete reverse remodeling groups. While no significant differences in UPS components were observed between these groups, there were significant correlations arising among various cardiac parameters and the UPS. Interestingly, total preoperative ubiquitination levels correlated with increased markers of hypertrophy, while individual E3 ligases had both positive and inverse correlations. Importantly, the authors directly address the limitations of this data set ie. Lack of postoperative protein levels. This data while correlative has the potential, with further study, to identify therapeutic targets and/or predict patient outcomes.

Comments:

1. Most of the data from table 3 is duplicated in figure 2. Should choose one or the other for publication.

2. Why are some non-significant correlations listed in table 3 and others are left blank?

6. PLOS authors have the option to publish the peer review history of their article (what does this mean?). If published, this will include your full peer review and any attached files.

Reviewer #1: No

---

## [Author Response · Author response to Decision Letter 0]

14 Jul 2020

Dear editors,

We are submitting the revised version of the manuscript entitled “Preoperative myocardial expression of E3 ubiquitin ligases in aortic stenosis patients undergoing valve replacement and their association to postoperative hypertrophy” for consideration by the PLOS ONE.

In this version we addressed the two points raised during the revision process, namely data duplication (figure 2 was eliminated, table 3 was maintained) and data display on table 3, which now depicts only significant (p < 0.05) and marginally significant (p ≤ 0.10) correlations.

Acknowledging the editorial instructions, we have revised the manuscript to comply with the requested formatting rules. Also, both Figure 1 and Supplementary Figure 1 are being resubmitted, after being uploaded to PACE and adjusted according to PLOS ONE criteria. Regarding the original blot images, these are being submitted as supporting information (“S1_raw_images”). Blot scans have been exported as TIFF images directly from Image Lab, and the requested annotations have been added post hoc on the PDF file. In such a file, beyond the five original scans, we have added the colorimetric scans used to estimate molecular weight from the protein ladder. These powerpoint slides, which had been already uploaded in the first submission, are now in the end of S1_raw_images file. 

We reiterate that the present “research article” is an original submission to PLOS ONE, which has not been published nor is it currently under consideration for publication elsewhere. Also, the authors declare no conflicts of interest. 

We hope that this manuscript fulfils the demanded quality for publication in PLOS ONE.

Looking forward to hearing from you.

Yours truthfully, Inês Falcão Pires

---

## [Editor Report · Decision Letter 1]

20 Jul 2020

Preoperative myocardial expression of E3 ubiquitin ligases in aortic stenosis patients undergoing valve replacement and their association to postoperative hypertrophy

PONE-D-20-03272R1

Dear Dr. Falcão-Pires,

We’re pleased to inform you that your manuscript has been judged scientifically suitable for publication and will be formally accepted for publication once it meets all outstanding technical requirements.

Kind regards,

Cécile Oury

Academic Editor

PLOS ONE
---

## [Editor Report · Acceptance letter]

10 Sep 2020

PONE-D-20-03272R1 

Preoperative myocardial expression of E3 ubiquitin ligases in aortic stenosis patients undergoing valve replacement and their association to postoperative hypertrophy 

Dear Dr. Falcão-Pires:

I'm pleased to inform you that your manuscript has been deemed suitable for publication in PLOS ONE. Congratulations! Your manuscript is now with our production department. 

Kind regards, 

on behalf of

Dr. Cécile Oury 

Academic Editor

PLOS ONE